# MODEL PARALLELISM WITH SUBNETWORK DATA PARALLELISM

## ABSTRACT

Pre-training large neural networks at scale imposes heavy memory demands on accelerators and often requires costly communication. We introduce Subnetwork Data Parallelism (SDP), a distributed training framework that partitions a model into structured subnetworks trained across workers without exchanging activations. We study two complementary masking regimes: *backward masking*, which applies sparsity only in the backward step to retain unbiased gradients, and *forward masking*, which also removes parameters in the forward pass to deliver stronger efficiency gains while providing additional regularization. We further explore two subnetwork construction strategies: *neuron level* and *block level*, applied across both CNNs and transformers. In experiments spanning CNNs and transformers on CIFAR and ImageNet, as well as LLM pre-training on FineWeb, SDP reduces per-device memory usage by **30%-75%** while maintaining or improving performance. Notably, in FLOP-matched settings, forward masking can sometimes achieve better performance.

## 1 INTRODUCTION

The rapid scaling of deep neural networks has led to unprecedented progress across a wide range of domains, from computer vision (He et al., 2016a; Radford et al., 2021; Oquab et al., 2023; Kirillov et al., 2023; Shang et al., 2024) to natural language processing (Bommasani et al., 2021; Achiam et al., 2023; Touvron et al., 2023; Zhao et al., 2023a). Training such large models has necessitated distributed strategies like *data parallelism* (Li et al., 2020) and *model parallelism* (Shazeer et al., 2018; Shoeybi et al., 2019; Huang et al., 2019), each with trade-offs. Data parallelism, typically implemented as Distributed Data Parallel (DDP) (Li et al., 2020), replicates the model on each GPU and synchronizes gradients after every backward pass. While simple and widely used, it incurs high memory overhead from full replication and high communication cost during synchronization. Model parallelism (e.g., GPipe (Huang et al., 2019)) mitigates memory issues by splitting layers across devices but requires expensive high-bandwidth interconnects to communicate activations. Unlike data parallelism, where several methods reduce communication cost (Douillard et al., 2023; Wang et al., 2023), lowering activation bandwidth remains an open challenge. Moreover, pipeline approaches often suffer inefficiencies from idle waiting (pipeline bubbles).

In this work, we propose *Subnetwork Data Parallelism* (SDP), a complementary strategy to model parallelism that reduces per-node memory by distributing the training of model sub-components across nodes. Unlike pipelining, which splits computation into sequential stages, SDP assigns each worker a *subnetwork*, a structurally complete portion of the model (e.g., removing rows and columns of a linear operator) that preserves a full path from input to loss, enabling independent gradient computation without exchanging activations. Each worker optimizes its subnetwork and synchronizes overlapping parameters through stepwise averaging.

We study two instantiations: (i) *forward-masked subnetworks*, which remove both forward and backward computation for a subnetwork, reducing parameters, activations, and gradients for substantial memory savings; and (ii) *backward-masked subnetworks*, where the forward pass uses the full model while masking is applied only in backpropagation, saving gradients and accumulators. The latter retains unbiased gradients and offers a theoretically grounded baseline, while the former provides a practical simplification that empirically improves stability and efficiency.

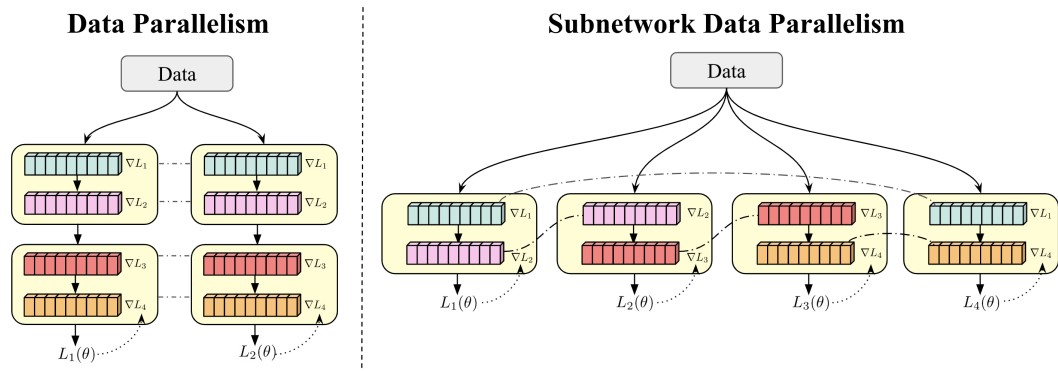

Figure 1: **Data Parallelism (DDP) vs. Subnetwork Data Parallelism (SDP).** *Left:* In data parallelism each GPU hosts a full replica, computes all layer gradients $\{\nabla L_1, \nabla L_2, \nabla L_3, \nabla L_4\}$, and all-reduces *all* parameters each step; per-GPU memory is approximately the full model (parameters + gradients + optimizer state + activations). *Right:* In SDP each GPU trains an end-to-end *subnetwork* (a subset of layers/neurons) with a local loss $L_k(\theta)$; only gradients of *shared* parameters are synchronized via masked averaging (dashed arcs). For a ***coverage ratio*** $\mathcal{C} = p/n$ (each parameter resides on $p$ of $n$ GPUs), both memory and communication per GPU scale as $\approx \mathcal{C} \times$ DP, with no cross-GPU activation exchange. This enables fitting larger models or longer sequences under the same hardware budget and improves scalability when bandwidth or memory are bottlenecks; when $\mathcal{C} = 1$ (all parameters on all GPUs), SDP reduces to standard DDP.

Rather than replicating or fully sharding the model, SDP distributes subnetworks across nodes so each device holds only a fraction of parameters (or gradients/accumulators for backward-masked). Subnetworks are trained independently and synchronized via parameter averaging, yielding a unified model. This significantly lowers memory usage while remaining compatible with intra-node data parallelism and existing systems-level model-parallel techniques.

Unlike pipelining, sharding, or tensor parallelism, our approach modifies the forward and backward computation. Its design rests on three observations. First, overlapping parameter assignments with periodic averaging maintain partial synchronization across workers; in forward masked subnetworks, each worker can be viewed as a replica constrained to remain similar through shared overlaps, akin to ensemble alignment strategies (Jolicoeur-Martineau et al., 2023a; Fournier et al., 2024). Second, in the backward masked regime, the forward pass uses the full model while sparsity is applied only during backpropagation. In this case, gradient estimates remain unbiased, and deviations from full DP are governed by mask connectivity, providing a principled baseline with theoretical guarantees. Third, subnetworks reduce per-iteration time thus decreased convergence rates (which we demonstrate theoretically) can be offset by increasing iterations in a FLOP-matched manner.

Our primary contributions in this work are:

- We propose a novel distributed training paradigm: ***Subnetwork Data Parallelism*** (SDP) enabling memory efficient distributed training. We provide a theoretical basis for our method linking the convergence in the backward masking case to a notion of a spectral gap.
- We explore two subnetwork construction strategies: (i) selecting subsets of neurons or channels, and (ii) removing entire layers (blocks) from the network, and compare them with standard Data Parallelism (DDP).
- We demonstrate that our approach achieves competitive performance on image classification tasks and large language model training, while substantially reducing per-device memory usage and synchronization overhead.

## 2 RELATED WORK

**Pipeline parallelism:** Pipeline parallelism reduces memory bottlenecks by splitting the model across devices. Huang et al. (2019); Rivaud et al. (2024) partition layers and pipeline micro-batches, while Mesh-TensorFlow (Shazeer et al., 2018) and Megatron-LM (Shoeybi et al., 2019) shard weights and activations within layers. These methods overcome memory limits but require high bandwidth

interconnects and still suffer from pipeline bubbles and load imbalance. Another line of work explores parallel layer training via auxiliary local losses (Belilovsky et al., 2020).

**Fully Sharded and Zero Redundancy Approaches.** To reduce memory inefficiencies in data parallelism, methods like Fully Sharded Data Parallel (FSDP) (Zhao et al., 2023b) and ZeRO (Rajbhandari et al., 2020) partition parameters, gradients, and optimizer states across devices. These approaches, supported by frameworks such as DeepSpeed (Rasley et al., 2020), greatly lower per-device memory but still incur substantial communication, especially during gradient synchronization, leading to higher overhead and latency.

**Ensemble Learning** Recent work (Fournier et al., 2024; Jolicoeur-Martineau et al., 2023b) shows the benefits of training multiple related models in parallel. Forward Subnetwork Masking can be seen as a similar framework, training diverse yet connected models while enforcing alignment, but with the added advantage of reduced per-iteration compute and memory.

**SWARM Learning** SWARM (Ryabinin et al., 2023) addresses model parallelism limits by assigning multiple devices to each pipeline stage and routing samples efficiently. In contrast, it still requires activation communication over potentially low-bandwidth links, whereas our subnetwork approach reduces all communication to parameters or gradients while maintaining only data parallelism across nodes.

**Federated Learning and Dropout-Based Subnetwork Training.** Federated learning frameworks (Konečný et al., 2016) train models across decentralized data sources, often addressing the non-iid challenge. Several works explore training subnetworks per device in this setting (Caldas et al., 2018; Horvath et al., 2021; Guliani et al., 2022; Wen et al., 2022; Alam et al., 2022), but with different goals and methodology. These focus on reducing communication and device compute, whereas our aim is to lower memory requirements, critical for training large models on memory limited GPUs. Communication load, by contrast, is well studied and can be mitigated through multi-step training and compression methods (Reddi et al., 2021; Douillard et al., 2023; Wang et al., 2023).

FedRolex and HeteroFL vary model size across clients to address device heterogeneity in compute and memory, assigning subnetworks via channel-level dropout. Our work instead targets a homogeneous setting, aiming to lower per-node memory through a more general subnetwork assignment strategy. Moreover, while these methods operate under privacy and heterogeneity constraints, often leaving each client with only small datasets, we assume each worker can access the full dataset. This avoids issues of heterogeneity or overfitting.

In these works, assigned masks are dynamic which adds significant communication and coordination overhead in a non-federated setting where wall-clock time is critical. Yuan et al. (2019) studied dynamic non-overlapping subnetworks with local SGD, whereas our fixed masks simplify the system and enable efficient forward and backward strategies. The overlapping nature of fixed subnetworks is key: shared assignments keep parameters aligned through averaging, and our analysis shows that convergence quality degrades with reduced overlap. Moreover, while Yuan et al. (2019) focused on MLPs, our method scales to standard architectures for image classification and large-scale language pre-training.

To the best of our knowledge, no related work in distributed subnetwork training considers masking only the backward pass, which retains many of the benefits of subnetworks. Fagnou et al. (2025) examined skipping backward blocks in residual networks to speed up training, but did not address distributed settings or memory reduction in the FLOP-matched regime.

## 3 METHOD

We introduce a distributed training framework that enhances memory efficiency in gradients, activation, and weight storage by defining a communication pattern between workers and model parameters. First we describe a generic multi-worker masking framework which considers fixed masks on parameters, gradients or both in the forward and backward pass of training. Then we specialize this to structured masks that yield benefits in memory and per-iteration speed.

## 3.1 FORWARD AND BACKWARD MASKING

**Gated coordinates.** Consider a distributed setting with $n$ workers (GPUs). Let $J$ be an index set of *coordinates* of the model; we use "coordinate" to refer to an index of the parameter vector $\boldsymbol{\theta}$ and, by the same index set, the corresponding coordinate of its gradient $\nabla_{\boldsymbol{\theta}}\mathcal{L}$. Each coordinate $j \in J$ is assigned to a subset of workers, with overlaps allowed. This assignment is encoded by a binary masking matrix $\mathbf{m} \in \{0,1\}^{n \times |J|}$, where $m_{i,j} = 1$ means worker $i$ is responsible for coordinate $j$. Using this mask, we define the *gated parameters* by elementwise multiplication with the global parameter vector:

$$\forall i \leq n, \forall j \in J, \quad (\mathbf{m} \odot \boldsymbol{\theta})_{i,j} \triangleq m_{i,j}\,\theta_j. \tag{gate}$$

Let $d_j \triangleq \sum_{i=1}^{n} m_{i,j}$ denote the degree of coordinate $j$; we assume $d_j \geq 1$ for all $j \in J$.

Given per–worker gradients $\mathbf{g}_1, \ldots, \mathbf{g}_n \in \mathbb{R}^{|J|}$, we define the *gated average* for $j \in J$ as the columnwise average over assigned workers:

$$\bar{\mathbf{m}}(\mathbf{g}_1, \ldots, \mathbf{g}_n)_j \triangleq \frac{1}{\sum_{i=1}^{n} \mathbf{m}_{i,j}} \sum_{i=1}^{n} \mathbf{m}_{i,j}(\mathbf{g}_i)_j = \frac{1}{d_j} \sum_{i=1}^{n} m_{i,j}(\mathbf{g}_i)_j. \tag{average}$$

We interpret $\mathbf{m}$ as the adjacency of a bipartite graph between coordinates $\{1, \ldots, |J|\}$ and workers $\{1, \ldots, n\}$. In practice, we require each coordinate to be assigned to at least one worker ($\min_j d_j \geq 1$) and encourage balanced worker degrees $\sum_j m_{i,j}$ to avoid load imbalance. In particular, this shows that $\bar{\mathbf{m}}(\mathbf{1}, \ldots, \mathbf{1}) = \mathbf{1}$. We therefore assume the corresponding bipartite graph is connected, which enforces agreement among the workers and is necessary for convergence (Nabli & Oyallon, 2023).

To compare masked averaging to the full-data case, let $\mathbf{m}^{\mathrm{uni}}$ denote the uniform assignment with $\mathbf{m}_{i,j}^{\mathrm{uni}} = 1$ for all $i \leq n, j \in J$. We view both averaging operators as acting columnwise; write $\|\cdot\|$ for the Euclidean norm on $\mathbb{R}^{|J|}$, and use $\sum_{i=1}^{n} \|\mathbf{g}_i\|^2$ to denote the squared Frobenius norm of the stacked gradients. Define the spectral gap

$$\rho \triangleq \sup_{\sum_{i=1}^{n} \|\mathbf{g}_i\|^2 \leq 1,\, \bar{\mathbf{m}}^{\mathrm{uni}}(\mathbf{g}_1,\ldots,\mathbf{g}_n)=0} \left\|\bar{\mathbf{m}}(\mathbf{g}_1, \ldots, \mathbf{g}_n)\right\|, \tag{1}$$

i.e., the largest singular value of $\bar{\mathbf{m}}$ restricted to the subspace orthogonal to the uniform direction.

**Proposition 1** (Deviation bound under backward masking). *Let $\rho \geq 0$ be the spectral gap defined above. Then for any collection of vectors $\mathbf{g}_1, \ldots, \mathbf{g}_n \in \mathbb{R}^{|J|}$,*

$$\left\|\bar{\mathbf{m}}^{\mathrm{uni}}(\mathbf{g}_1, \ldots, \mathbf{g}_n) - \bar{\mathbf{m}}(\mathbf{g}_1, \ldots, \mathbf{g}_n)\right\|^2 \leq \rho^2 \sum_{i=1}^{n} \|\mathbf{g}_i\|^2. \tag{2}$$

**Forward and Backward Masking** We assume that we have access to two masks $\mathbf{m}_{\mathrm{fwd}}, \mathbf{m}_{\mathrm{bwd}}$. At step $t$, worker $i$ draws a mini-batch $\mathcal{B}_i^{(t)}$ using a forward mask $\mathbf{m}_{\mathrm{fwd}}$. The forward pass evaluates the loss at

$$\boldsymbol{\theta}_i^{(t)} = (\mathbf{m}_{\mathrm{fwd}} \odot \boldsymbol{\theta}^{(t)})_i, \qquad \mathbf{g}_i^{(t)} = \nabla_{\boldsymbol{\theta}}\mathcal{L}\left(\boldsymbol{\theta}_i^{(t)}; \mathcal{B}_i^{(t)}\right).$$

Note at this stage, that if $(\mathbf{m}_{\mathrm{fwd}})_{i,j} = 0$ then $(\mathbf{g}_i^{(t)})_j = 0$. The backward pass applies the backward aggregation mask on the resulting gradients $\mathbf{m}_{\mathrm{bwd}}$ componentwise:

$$\hat{\mathbf{g}}^{(t)} = \bar{\mathbf{m}}_{\mathrm{bwd}}\left(\mathbf{g}_1^{(t)}, \ldots, \mathbf{g}_n^{(t)}\right),$$

followed by the optimizer update

$$\boldsymbol{\theta}^{(t+1)} = \boldsymbol{\theta}^{(t)} - \mathrm{OptUpdate}\left(\hat{\mathbf{g}}^{(t)}, \mathbf{s}\right).$$

We consider two variants of this under the Subnetwork DP framework

- *Forward-masking:* $\mathbf{m}_{\mathrm{fwd}} = \mathbf{m}$ and $\mathbf{m}_{\mathrm{bwd}} = \mathbf{m}$. The model is evaluated at masked parameters; activations are gated and memory-saving, but gradients reflect this masked forward.

- *Backward-masking:* $\mathbf{m}_{\mathrm{fwd}} = \mathbf{m}^{\mathrm{uni}}$ and $\mathbf{m}_{\mathrm{bwd}} = \mathbf{m}$. The model is evaluated at full $\boldsymbol{\theta}^{(t)}$ (no forward bias); sparsity appears only in backprop/aggregation.

Choosing $\mathbf{m}_{\mathrm{fwd}} = \mathbf{m}^{\mathrm{uni}}$ keeps activations identical across workers and removes deviation due to masked aggregation governed by the spectral gap $\rho$.

**Deviation under Backward-masking.** With Backward-masking, define

$$\widehat{\mathbf{g}}^{(t)} = \bar{\mathbf{m}}\big(\mathbf{g}_1^{(t)}, \dots, \mathbf{g}_n^{(t)}\big), \qquad \mathbf{g}_{\mathrm{uni}}^{(t)} = \bar{\mathbf{m}}^{\mathrm{uni}}\big(\mathbf{g}_1^{(t)}, \dots, \mathbf{g}_n^{(t)}\big).$$

By Proposition 2,

$$\big\|\widehat{\mathbf{g}}^{(t)} - \mathbf{g}_{\mathrm{uni}}^{(t)}\big\|^2 \;\leq\; \rho^2 \sum_{i=1}^{n} \|\mathbf{g}_i^{(t)}\|^2. \tag{3}$$

**Convergence in the $L$-smooth case (Backward-masking, simple).** Assume $f : \mathbb{R}^{|J|} \to \mathbb{R}$ is $L$-smooth. In the backward-masked (BM) setting we take

$$\mathbf{m}_{\mathrm{fwd}} = \mathbf{m}^{\mathrm{uni}}, \qquad \mathbf{m}_{\mathrm{bwd}} = \mathbf{m},$$

so the forward pass is unmasked and only the backward/aggregation is masked. Each worker $i$ computes a stochastic gradient $\mathbf{g}_i^{(t)}$ at $\boldsymbol{\theta}^{(t)}$ with

$$\mathbb{E}\Big[\mathbf{g}_i^{(t)} \mid \boldsymbol{\theta}^{(t)}\Big] = \nabla f(\boldsymbol{\theta}^{(t)}), \qquad \mathbb{E}\Big[\|\mathbf{g}_i^{(t)} - \nabla f(\boldsymbol{\theta}^{(t)})\|^2 \mid \boldsymbol{\theta}^{(t)}\Big] \leq \sigma^2,$$

and we aggregate by masked averaging

$$\widehat{\mathbf{g}}^{(t)} \;=\; \bar{\mathbf{m}}\Big(\mathbf{g}_1^{(t)}, \dots, \mathbf{g}_n^{(t)}\Big).$$

Let $\mathbf{g}_{\mathrm{uni}}^{(t)} = \bar{\mathbf{m}}^{\mathrm{uni}}(\mathbf{g}_1^{(t)}, \dots, \mathbf{g}_n^{(t)})$ and define the masking error $\boldsymbol{\delta}^{(t)} \triangleq \widehat{\mathbf{g}}^{(t)} - \mathbf{g}_{\mathrm{uni}}^{(t)}$. By linearity of expectation, both $\widehat{\mathbf{g}}^{(t)}$ and $\mathbf{g}_{\mathrm{uni}}^{(t)}$ are unbiased for $\nabla f(\boldsymbol{\theta}^{(t)})$ under BM. The update is

$$\boldsymbol{\theta}^{(t+1)} = \boldsymbol{\theta}^{(t)} - \eta \widehat{\mathbf{g}}^{(t)}.$$

**Theorem 1** (SGD rate under Backward-masking). *If $f$ is $L$-smooth and $\eta \leq \frac{1}{2L(1+n\rho^2)}$, then for any $T \geq 1$,*

$$\frac{1}{T} \sum_{t=0}^{T-1} \mathbb{E}\left\|\nabla f(\boldsymbol{\theta}^{(t)})\right\|^2 \;\leq\; \frac{2\big(f(\boldsymbol{\theta}^{(0)}) - f^\star\big)}{\eta T} \;+\; 2L\eta\left(\frac{\sigma^2}{n} + n\rho^2\sigma^2\right),$$

*where $f^\star \triangleq \inf_{\boldsymbol{\theta}} f(\boldsymbol{\theta})$ and $\rho$ is the spectral gap of the masking operator.*

## 3.2 Subnetwork Data Parallelism with Structured Mask Construction

Our framework instantiates *Subnetwork Data Parallelism (SDP)* by employing *structured masks*, which remove entire parameter groups including parameters, gradients, accumulators, and activations from each worker. This yields substantial memory savings and per-iteration speedups, offsetting slower convergence while preserving the efficiency benefits of subnetworks. We introduce two strategies for instantiating *subnetworks*: *Neuron-Level SDP (N-SDP)*, based on dropout (Srivastava et al., 2014) for fully connected and convolutional layers, and *Block-Level SDP (B-SDP)*, inspired by stochastic depth (Huang et al., 2016) for residual architectures.

**Neuron-Level SDP (N-SDP).** Through **N-SDP** we instantiate subnetworks by selectively removing neurons in fully connected layers (or channels in convolutional layers). For two successive layers $(W^l, W^{l+1})$ with $W^l : \mathbb{R}^{d_{l-1}} \to \mathbb{R}^{d_l}$, dropping outputs of layer $l$ naturally removes the corresponding inputs of layer $l+1$. For simplicity, we restrict to *forward masking*, where the same mask is applied in both directions ($\mathbf{m}_{\mathrm{fwd}} = \mathbf{m}_{\mathrm{bwd}} = \mathbf{m}$; see Section 3.1). Applying $m^l$ to layer $l$ thus induces a consistent $m^{l+1}$ on layer $l+1$. As a result,

$$(\mathbf{m}^l \odot W^l, \, W^{l+1}) \quad \text{and} \quad (\mathbf{m}^l \odot W^l, \, \mathbf{m}^{l+1} \odot W^{l+1})$$

produce identical outputs. For example, if $W^l, W^{l+1} \in \mathbb{R}^{d \times d}$ and we mask a subset $J_{\mathrm{mask}} \subset \{1, \dots, d\}$ of output neurons, then setting

$$m_{jk}^l = 0, \quad j \in J_{\mathrm{mask}}, \, k \in \{1, \dots, d\}, \qquad m_{kj}^{l+1} = 0, \quad j \in J_{\mathrm{mask}}, \, k \in \{1, \dots, d\},$$

ensures that both layers remain consistent under the masking operation.

**Block-Level SDP (B-SDP).** Here, subnetworks are formed by removing entire blocks, particularly in architectures with skip connections. Let the model have $L$ blocks $\{B^1, \ldots, B^L\}$ with parameters $\boldsymbol{\theta}^{(l)}$. Each block has a binary mask $m^{(l)} \in \{0, 1\}$ denoting whether it is active. When $m^{(l)} = 0$, the block is skipped and its parameters excluded. In residual architectures (e.g., ResNets), this reduces to the identity mapping via the skip path, ensuring valid representations even when blocks are dropped. Formally, for a residual connection of the form

$$B^{(l)}(\mathbf{x}) + \mathbf{x},$$

the masked computation at block $l$ is

$$\hat{B}^{(l)}(\mathbf{x}) = m^{(l)} B^{(l)}(\mathbf{x}) + \mathbf{x}. \tag{4}$$

We also consider the more general case of *backward masking*, where $\mathbf{m}_{\text{fwd}} = \mathbf{m}^{\text{uni}}$ and $\mathbf{m}_{\text{bwd}} = \mathbf{m}$ as explained in Section 3.1. We refer this instantiation as $\mathbf{B}_b$**-SDP** where the block may be active during the forward pass but omitted during the backward pass.

**Memory, compute, and communication cost** Let $N$ be the total parameter count and $\mathcal{C} \in (0, 1]$ the per-worker density (fraction of coordinates selected by the mask). Consider bf16 parameters (2 bytes), fp32 gradients (4 bytes), and Adam accumulators in fp32 (8 bytes), standard DP requires $\approx 14N$ bytes per-worker. In *Forward masking:* only the $\mathcal{C}N$ coordinates materialize parameters, gradients, and accumulators, using $\approx 14\mathcal{C}N$ bytes; activations and compute also scale $\approx \mathcal{C}$ for structured masks (channel/block level). In *Backward masking:* the full forward is computed, but gradients and accumulators are stored only for the $\mathcal{C}N$ active coordinates, giving $(2 + 12\mathcal{C})N$ bytes. Activations scale $\approx \mathcal{C}$. Block forward masking illustrated in Figure 1 and compared with DDP pipelining.

**Communication cost** Communication cost is also reduced under SDP. In ring all-reduce, each worker with $N$ parameters sends and receives about $2N$ scalars per step, whereas SDP synchronizes only the $\mathcal{C}N$ active coordinates, reducing the cost to $\approx 2\mathcal{C}N$. When masks differ across workers, each parameter block is reduced only within its subset of workers; this holds for both forward and backward masking. Gradient compression schemes are well studied in data-parallel settings (Shi et al., 2019; Xu et al., 2021), offering additional savings, but efficient activation compression (e.g., in pipelining or tensor parallelism) remains poorly understood. Thus, SDP can sometimes operate where bandwidth limits preclude other model-parallel methods, while standard techniques (tensor, sharding, pipelining, context) can still be applied within each SDP replica to further reduce memory for large models.

## 4 EXPERIMENTS

We now describe our experimental setup for our proposed ***Subnetwork Data Parallelism*** (SDP) framework on a number of tasks including CIFAR-10 / CIFAR-100 (Krizhevsky et al., 2009), ImageNet (Deng et al., 2009; Russakovsky et al., 2015), and LLM training on FineWeb dataset (Penedo et al., 2024). We shall release code for reproducibility at the time of publication.

For **N-SDP** we define ***coverage ratio*** ($\mathcal{C}$) as $p/n$, where $p$ denotes the number of active workers (out of the total $n$ workers/GPUs) that share a given parameter $\theta_j$. This overlap quantifies the sparsity with which each subnetwork is trained across the $n$ workers. For example, an overlap of $p/n = 6/8$ means that for every parameter $\theta_j$ in the parameter vector $\theta$, exactly 6 of the 8 workers participate in its training. By contrast, $p/n = 8/8 = 1$ corresponds to the standard data-parallel (DP) setup, where all workers contribute to the training of every parameter.

Similarly for **B-SDP** and $\mathbf{B}_b$**-SDP** we define ***active blocks*** ($\mathcal{A}$) as $b/d$, where $b$ denotes the number of active computational blocks (for example Basic Block in ResNets and Attention+MLP Block in Transformers) assigned to each worker out of the total $d$ computational blocks in the model.

In all cases we either use standard hyperparameters for the task from the literature or tune the hyperparameters on DDP (e.g. for LLM experiments) and reuse the same on all SDP settings. We note that tuning hyperparameters for SDP can be a practical approach to further improve performance in practice.

Table 1: Top-1 test accuracy (%) (↑) with **RN-18 and WRN-18** using a cosine annealing scheduler across different coverage ratios ($\mathcal{C}$) comparing **N-SDP**, **B-SDP**, and **B$_b$-SDP** with standard DDP ($\mathcal{C}=1$). Blue cells match or exceed DDP within error bars with $\mathcal{C}=5/8$ giving 37.5% memory savings, while at extreme sparsity ($\mathcal{C}=3/8$) **B$_b$-SDP** avoids performance collapse.

| | | | | ResNet-18 (RN-18) | | | |
|---|---|---|---|---|---|---|---|
| **Dataset** | **Masking** | **DDP** ($\mathcal{C}=1$) | $\mathcal{C}=7/8$ | $\mathcal{C}=6/8$ | $\mathcal{C}=5/8$ | $\mathcal{C}=4/8$ | $\mathcal{C}=3/8$ |
| CIFAR-10 | **N-SDP** | | $92.81_{\pm 0.23}$ | $92.72_{\pm 0.23}$ | $92.49_{\pm 0.09}$ | $91.47_{\pm 0.29}$ | $22.56_{\pm 2.04}$ |
| | **B-SDP** | $92.45_{\pm 0.14}$ | $\mathbf{93.18}_{\pm 0.16}$ | $92.89_{\pm 0.18}$ | $89.52_{\pm 0.16}$ | $84.72_{\pm 0.40}$ | $42.68_{\pm 2.09}$ |
| | **B$_b$-SDP** | | $92.14_{\pm 0.14}$ | $91.33_{\pm 0.02}$ | $90.24_{\pm 0.04}$ | $88.80_{\pm 0.11}$ | $87.91_{\pm 0.29}$ |
| CIFAR-100 | **N-SDP** | | $69.02_{\pm 0.14}$ | $68.42_{\pm 0.35}$ | $67.69_{\pm 0.59}$ | $65.20_{\pm 0.12}$ | $9.79_{\pm 2.51}$ |
| | **B-SDP** | $68.62_{\pm 0.01}$ | $\mathbf{70.14}_{\pm 0.48}$ | $68.84_{\pm 0.28}$ | $54.27_{\pm 0.51}$ | $36.20_{\pm 0.01}$ | $7.03_{\pm 0.40}$ |
| | **B$_b$-SDP** | | $67.33_{\pm 0.43}$ | $64.90_{\pm 0.24}$ | $61.87_{\pm 0.16}$ | $59.89_{\pm 0.35}$ | $58.73_{\pm 0.39}$ |
| | | | | WideResNet-18 (WRN-18) | | | |
| **Dataset** | **Masking** | **DDP** ($\mathcal{C}=1$) | $\mathcal{C}=7/8$ | $\mathcal{C}=6/8$ | $\mathcal{C}=5/8$ | $\mathcal{C}=4/8$ | $\mathcal{C}=3/8$ |
| CIFAR-10 | **N-SDP** | | $93.44_{\pm 0.03}$ | $93.33_{\pm 0.04}$ | $93.36_{\pm 0.21}$ | $92.98_{\pm 0.09}$ | $55.34_{\pm 5.65}$ |
| | **B-SDP** | $93.01_{\pm 0.08}$ | $\mathbf{93.78}_{\pm 0.07}$ | $93.61_{\pm 0.01}$ | $91.54_{\pm 0.16}$ | $88.34_{\pm 0.15}$ | $58.06_{\pm 8.39}$ |
| | **B$_b$-SDP** | | $92.65_{\pm 0.14}$ | $92.07_{\pm 0.10}$ | $91.24_{\pm 0.14}$ | $89.87_{\pm 0.43}$ | $88.28_{\pm 0.61}$ |
| CIFAR-100 | **N-SDP** | | $68.80_{\pm 0.75}$ | $69.14_{\pm 0.11}$ | $68.96_{\pm 0.38}$ | $68.24_{\pm 0.03}$ | $44.74_{\pm 0.47}$ |
| | **B-SDP** | $69.12_{\pm 0.41}$ | $\mathbf{70.97}_{\pm 0.41}$ | $68.27_{\pm 0.12}$ | $56.90_{\pm 0.31}$ | $42.23_{\pm 0.85}$ | $9.25_{\pm 0.53}$ |
| | **B$_b$-SDP** | | $67.51_{\pm 0.28}$ | $65.37_{\pm 0.46}$ | $62.82_{\pm 0.14}$ | $61.05_{\pm 0.31}$ | $59.91_{\pm 0.29}$ |

## 4.1 SDP WITH IMAGE CLASSIFICATION

### 4.1.1 RESNET-18 CNN ARCHITECTURE

**Experimental Setup:** We conduct experiments using ResNet-18 (He et al., 2016b) and its wider variant (Zagoruyko & Komodakis, 2016). We evaluate three *Subnetwork Data Parallel* (SDP) strategies: **N-SDP B-SDP** and **B$_b$-SDP**, as described in Section 3.2. Experiments are performed under different $\mathcal{C}$ by varying $p \in \{7, 6, 5, 4, 3\}$ on $n = 8$ GPUs. Since the ResNet-18 architecture contains $d = 8$ computational blocks, we similarly vary $\mathcal{A}$ as $b \in \{7, 6, 5, 4, 3\}$. All experiments with the ResNet-18 architecture are trained with standard hyperparameters (Zhuang et al., 2022; Cho et al., 2025): an effective batch size of $\mathcal{B} = 512$, corresponding to 64 samples per GPU across $n = 8$ workers with a cosine annealing learning rate schedule for 200 epochs. Further details regarding hyperparameters are given in Appendix B.

To ensure fair comparison, we FLOP-match all configurations by extending the target schedule and proportionally the warmup. For **N-SDP** and **B-SDP**, this is done by scaling training epochs inversely with the number of active parameters (e.g., $\mathcal{C} = 4/8$ doubles the schedule). For **B$_b$-SDP**, we account for the higher backward cost, so the same setting ($\mathcal{A} = 4/8$) increases training iterations by $1.5\times$.

Table 1 highlights the benefits of our proposed *Subnetwork Data Parallelism*. The primary advantage lies in reducing per-worker memory while retaining competitive accuracy. For example, with only 87.5% of parameters ($\mathcal{C} = 7/8$), both RN-18 and WRN-18 match or even surpass standard data parallelism (DDP) on CIFAR-10 and CIFAR-100, suggesting a regularization effect from subnetwork training in the forward masking case. Even at 50% parameters ($\mathcal{C} = 4/8$), performance remains competitive, particularly for **N-SDP**, while offering substantial memory savings. Across both models and datasets, performance degrades gracefully under reduced overlap, with WRN-18 showing greater robustness than RN-18 at high sparsity. Under severe sparsity ($\mathcal{C} = 3/8$), **B$_b$-SDP** clearly outperforms **N-SDP** and **B-SDP**, reaching $87.91\%$ on CIFAR-10 and $58.73\%$ on CIFAR-100, while the others collapse. Finally, repeating experiments with a linear scheduler yields consistent trends (Appendix E), underscoring the robustness of our framework.

Across both ResNet variants, **N-SDP** remains stable down to ($\mathcal{C} = 5/8$), while **B-SDP** degrades earlier. **B$_b$-SDP** shows gradual decline but surpasses **N-SDP** under extreme sparsity, notably at ($\mathcal{C} = 3/8$). This advantage of backward masking at very low overlap is consistent with the observation in Sec 3 that it maintains an unbiased gradient estimate (but at a slower iteration level convergence). We attribute the advantage of forward masking at higher overlap to two factors: (1) it effectively trains multiple models in parallel, whose diversity improves performance when averaged (Fournier et al.,

Table 2: Top-1 test accuracy (%)(↑) with **Swin-T (Tiny)** across active blocks ($\mathcal{A}$) comparing **B-SDP** and **B$_b$-SDP** with DDP ($\mathcal{A}=1$). Blue cells match or exceed DDP within error bars with $\mathcal{A}=8/12$ giving 33.3% memory savings, while at extreme sparsity ($\mathcal{A}=5/12$) **B$_b$-SDP** avoids performance collapse.

| Dataset | Masking | DDP ($\mathcal{A}=1$) | $\mathcal{A}=10/12$ | $\mathcal{A}=9/12$ | $\mathcal{A}=8/12$ | $\mathcal{A}=6/12$ | $\mathcal{A}=5/12$ |
|---|---|---|---|---|---|---|---|
| CIFAR-10 | **B-SDP** | 90.66 $_{\pm0.01}$ | 90.92 $_{\pm0.11}$ | 90.63 $_{\pm0.16}$ | 90.22 $_{\pm0.04}$ | 86.86 $_{\pm0.18}$ | 73.12 $_{\pm0.96}$ |
| | **B$_b$-SDP** | | 89.90 $_{\pm0.16}$ | 89.14 $_{\pm0.17}$ | 88.05 $_{\pm0.17}$ | 85.56 $_{\pm0.12}$ | 83.14 $_{\pm0.14}$ |
| CIFAR-100 | **B-SDP** | 64.76 $_{\pm0.28}$ | 66.64 $_{\pm0.31}$ | 66.24 $_{\pm0.04}$ | 65.35 $_{\pm0.53}$ | 50.15 $_{\pm1.29}$ | 14.99 $_{\pm0.05}$ |
| | **B$_b$-SDP** | | 64.41 $_{\pm0.27}$ | 64.10 $_{\pm0.19}$ | 63.03 $_{\pm0.10}$ | 60.14 $_{\pm0.38}$ | 58.10 $_{\pm0.68}$ |

Table 3: Top-1 test accuracy (%) (↑) with **Swin-T (Tiny) on ImageNet-1k** across active blocks ($\mathcal{A}=11/12$) comparing **B-SDP** and **B$_b$-SDP** with DDP ($\mathcal{A}=1$). Performance degradation is greater in **B$_b$-SDP** as compared to that in Block Masking **B-SDP**.

| Dataset | Masking | DDP ($\mathcal{A}=1$) | $\mathcal{A}=11/12$ |
|---|---|---|---|
| ImageNet | **B-SDP** | 81.01 | 79.30 |
| | **B$_b$-SDP** | | 77.78 |

2024; Jolicoeur-Martineau et al., 2023b; Douillard et al.); and (2) under FLOP matching, forward masking gains more training iterations than backward masking.

### 4.1.2 SDP WITH SWIN TRANSFORMER ARCHITECTURE

Motivated by the simplicity of **B-SDP** and **B$_b$-SDP** and the strong performance of **B$_b$-SDP** at low $\mathcal{C}$ (as seen in Table 1), we compare them against the DDP baseline. We train Swin-T (Tiny) (Liu et al., 2021) with $d=12$ transformer blocks and evaluate subnetworks on CIFAR-10/100 by varying $b \in \{10, 9, 8, 6, 5\}$, with an effective batch size of $\mathcal{B}=512$ across $n=8$ workers. Further details regarding hyperparameters are given in Appendix C.

Table 2 reports Swin-T results with **B-SDP** and **B$_b$-SDP** on CIFAR-10/100. As in ResNet, performance stays stable for Swin-T as active blocks decrease: on CIFAR-10, accuracy remains near 90% down to $\mathcal{A}=8/12$, dropping only beyond this (e.g., 86.86% at $\mathcal{A}=6/12$). On CIFAR-100, accuracy even improves by **2%**, from 64.76% (12 blocks) to 66.64% (10 blocks). We also perform experiments on ImageNet (Table 3) which follows the same hyper parameters as Liu et al. (2021), at $\mathcal{A}=11/12$, **B-SDP** achieves 79.30% versus 77.78% for **B$_b$-SDP**, compared to the 81.01% baseline. We note that standard ImageNet training for Swin is in a long training regime of 300 epochs that may be well beyond compute-optimal, we hypothesize that the long schedule used in the ImageNet training results lead to saturation in performance and thus a larger number of iterations would be needed for **B$_b$-SDP** to fully converge. In the subsequent section we show that the method can scale to standard large training settings used in LLMs when compared to a compute optimal training regime.

Table 4: Validation loss (Val. Loss), Perplexity (PPL), and Relative memory (Rel-Mem) normalized to the DDP Baseline memory $M$ for different $\mathcal{A}$. Overlaps in light blue indicate Val. Loss $\leq$ DDP Baseline (and the corresponding perplexity), with $\mathcal{A}=3/12$ giving 75% memory savings

| Metric | Masking | DDP ($\mathcal{A}=1$) | $\mathcal{A}=10/12$ | $\mathcal{A}=8/12$ | $\mathcal{A}=6/12$ | $\mathcal{A}=5/12$ | $\mathcal{A}=4/12$ | $\mathcal{A}=3/12$ |
|---|---|---|---|---|---|---|---|---|
| Val. Loss (↓) | **B-SDP** | 3.57 | 3.45 | **3.41** | 3.43 | 3.43 | 3.62 | 3.86 |
| | **B$_b$-SDP** | | 3.47 | 3.45 | 3.45 | 3.46 | 3.48 | 3.54 |
| PPL (↓) | **B-SDP** | 35.4 | 31.5 | **30.4** | 30.8 | 30.9 | 37.4 | 47.3 |
| | **B$_b$-SDP** | | 32.0 | 31.5 | 31.5 | 31.9 | 32.4 | 34.5 |
| Rel-Mem | **B-SDP** | $M$ | 0.83M | 0.67M | 0.50M | 0.42M | 0.33M | 0.25M |
| | **B$_b$-SDP** | | 0.87M | 0.73M | 0.60M | 0.53M | 0.47M | 0.40M |

### 4.2 SDP WITH LARGE LANGUAGE MODELS (LLMS)

We evaluated SDP on a 134M LLaMA-style model (Grattafiori et al., 2024), trained with a 3B-token budget (according to the Chinchilla scaling laws (Hoffmann et al., 2022) for the DDP baseline) on the FineWeb dataset. Hyperparameters are reported in Appendix D. LLMs tend to have significant

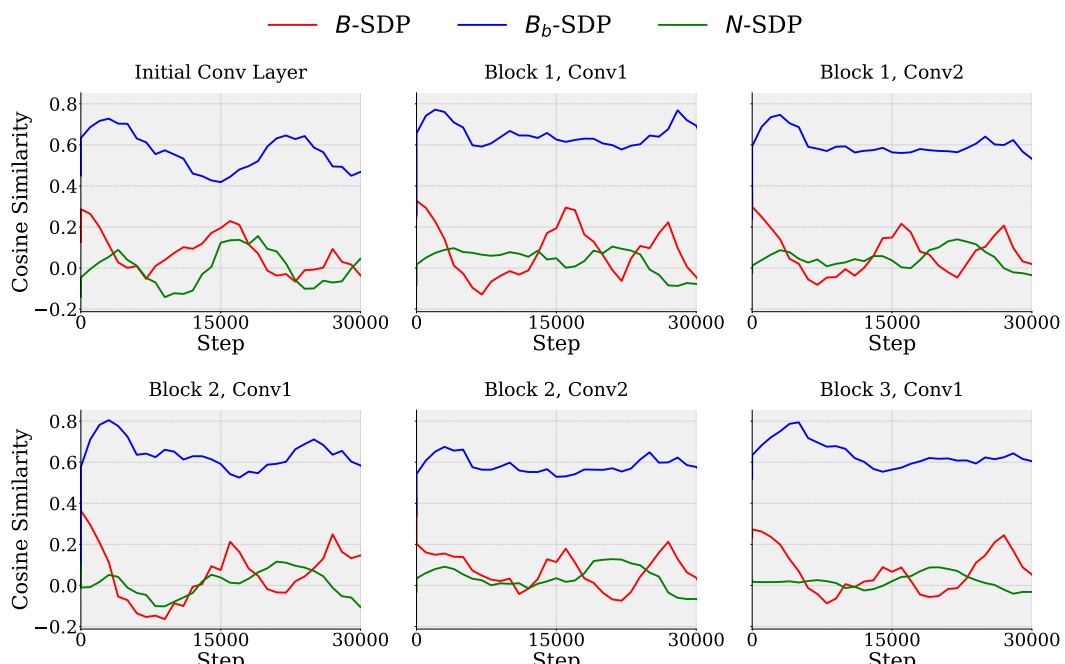

Figure 2: Cosine similarity between subnetworks with **N-SDP**, **B-SDP**, $\mathbf{B}_b$**-SDP** and a full ResNet-18 model's gradients, across various convolutional layers. The subnetworks constructed above have a *coverage ratio* ($\mathcal{C}= 4/8$) for **N-SDP** and same *active blocks* ($\mathcal{A}= 4/8$) for **B-SDP** and $\mathbf{B}_b$**-SDP**.

memory constraints in practice and thus SDP is a highly pertinent direction for reducing the per-node requirements. Our results are reported in Table 4. We compare **B-SDP** and $\mathbf{B}_b$**-SDP** with standard DDP, running all setups in a FLOP-matched regime. Subnetwork Data Parallelism (SDP) consistently outperforms DDP in this setting. **B-SDP** with $\mathcal{A}= 8/12$ achieves the best results outperforming DDP, with lowest validation loss (3.41 vs. 3.57) and perplexity (PPL) (30.4 vs. 35.4) while using only $\mathbf{0.67M}$ memory relative to the full DDP baseline $M$. At extreme sparsity ($\mathcal{A}= 3/12$), **B-SDP** degrades sharply, whereas $\mathbf{B}_b$**-SDP** remains stable and even surpasses DDP (Val. Loss = $\mathbf{3.54}$, PPL = $\mathbf{34.5}$). The results follow a similar pattern to our results on CIFAR-10: demonstrating both **B-SDP** and $\mathbf{B}_b$**-SDP** can be effective at higher overlap with **B-SDP** actually improving performance while $\mathbf{B}_b$**-SDP** showing better results than **B-SDP** at lower overlap.

### 4.3 QUANTITATIVE ANALYSIS

We study gradient alignment between the gradient that would be computed by the full model replica and the gradient produced by **N-SDP**, **B-SDP**, $\mathbf{B}_b$**-SDP**. For **B-SDP** and $\mathbf{B}_b$**-SDP**, only active blocks are compared; for **N-SDP**, only active parameters within each layer. Figure 2 shows alignment for ResNet-18 ($\mathcal{C}= 4/8$) across convolutional layers. $\mathbf{B}_b$**-SDP** maintains the highest cosine similarity ($\approx 0.6$) with the full model's gradients, aligning with expectation that restricting modifications to the backward pass better still leads to unbiased gradient estimates as discussed in Section 3.1. By contrast, **B-SDP** and **N-SDP** show near-zero similarity, indicating stronger divergence from full-model gradients, especially in early layers. Notably despite poor alignment the performance of the models still does not collapse (e.g. is above 91% for **N-SDP**).

## 5 CONCLUSION

In this work we present a novel distributed training framework: ***Subnetwork Data Parallelism*** (SDP) that delivers **30%–75%** memory savings per device while maintaining or even improving accuracy over DDP. By combining forward and backward masking with structured subnetwork construction, SDP scales gracefully across CNNs, transformers, and LLM pre-training. These results highlight SDP as a practical path toward training larger models under limited memory budgets.

## 6 REPRODUCIBILITY STATEMENT

We have taken several steps to ensure the reproducibility of our work. In Section 4 we describe the exact models, datasets and hyperparameters used. Our exact codebase will be released at the time of publication. In addition, the main text and Appendix B, Appendix C and Appendix D include all relevant details and a description of our hyperparameter tuning procedures, ensuring that our experiments can be fully reproduced.

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

## A  THEORETICAL ANALYSIS

**Proposition 2** (Deviation bound under masking)**.** *Let $\rho \geq 0$ be the spectral gap defined above. Then for any collection of vectors $\mathbf{g}_1, \ldots, \mathbf{g}_n \in \mathbb{R}^{|J|}$,*

$$\left\| \bar{\mathbf{m}}^{\mathrm{uni}}(\mathbf{g}_1, \ldots, \mathbf{g}_n) - \bar{\mathbf{m}}(\mathbf{g}_1, \ldots, \mathbf{g}_n) \right\|^2 \leq \rho^2 \sum_{i=1}^{n} \|\mathbf{g}_i\|^2. \tag{5}$$

*Proof.* Assume $|J|=1$ (the general case follows by summing the per–coordinate bounds). In this case, the averaging operators are linear forms $\mathbb{R}^n \to \mathbb{R}$ with $\bar{\mathbf{m}}(1, \ldots, 1) = \bar{\mathbf{m}}^{\mathrm{uni}}(1, \ldots, 1) = 1$. For any $v \in \mathbb{R}^n$, write $v = \alpha \mathbf{1} + u$ with $u \perp \mathbf{1}$. Then $(\bar{\mathbf{m}} - \bar{\mathbf{m}}^{\mathrm{uni}})v = \bar{\mathbf{m}}(u)$ since $\bar{\mathbf{m}}^{\mathrm{uni}}(u) = 0$. By definition of $\rho$, $|\bar{\mathbf{m}}(u)| \leq \rho \|u\| \leq \rho \|v\|$, which gives the claim after squaring. $\square$

**Theorem 2** (Nonconvex rate under BM)**.** *If $f$ is $L$-smooth and $\eta \leq \frac{1}{2L(1+n\rho^2)}$, then for any $T \geq 1$,*

$$\frac{1}{T} \sum_{t=0}^{T-1} \mathbb{E} \left\| \nabla f(\boldsymbol{\theta}^{(t)}) \right\|^2 \leq \frac{2\left( f(\boldsymbol{\theta}^{(0)}) - f^\star \right)}{\eta T} + 2L\eta \left( \frac{\sigma^2}{n} + n\rho^2\sigma^2 \right),$$

*where $f^\star \triangleq \inf_{\boldsymbol{\theta}} f(\boldsymbol{\theta})$ and $\rho$ is the spectral gap of the masking operator.*

*Proof (two steps). (1) Descent lemma.* By $L$-smoothness,

$$f(\theta^{t+1}) \leq f(\theta^t) - \eta \langle \nabla f(\theta^t), \widehat{\mathbf{g}}^{(t)} \rangle + \frac{L\eta^2}{2} \|\widehat{\mathbf{g}}^{(t)}\|^2.$$

Taking $\mathbb{E}[\cdot \mid \theta^t]$ and using $\mathbb{E}[\widehat{\mathbf{g}}^{(t)} \mid \theta^t] = \nabla f(\theta^t)$,

$$\mathbb{E}[f(\theta^{t+1}) \mid \theta^t] \leq f(\theta^t) - \eta \|\nabla f(\theta^t)\|^2 + \frac{L\eta^2}{2} \mathbb{E}\|\widehat{\mathbf{g}}^{(t)}\|^2.$$

*(2) Second moment of the masked estimator.* Decompose $\widehat{\mathbf{g}}^{(t)} = \mathbf{g}_{\mathrm{uni}}^{(t)} + \boldsymbol{\delta}^{(t)}$. Then

$$\mathbb{E}\|\widehat{\mathbf{g}}^{(t)}\|^2 \leq 2\,\mathbb{E}\|\mathbf{g}_{\mathrm{uni}}^{(t)}\|^2 + 2\,\mathbb{E}\|\boldsymbol{\delta}^{(t)}\|^2.$$

Unbiasedness gives $\mathbb{E}\|\mathbf{g}_{\mathrm{uni}}^{(t)}\|^2 \leq \|\nabla f(\theta^t)\|^2 + \sigma^2/n$. For the masking term, Proposition 2 yields

$$\mathbb{E}\|\boldsymbol{\delta}^{(t)}\|^2 \leq \rho^2 \sum_{i=1}^{n} \mathbb{E}\|\mathbf{g}_i^{(t)}\|^2 \leq \rho^2 \sum_{i=1}^{n} \left( \|\nabla f(\theta^t)\|^2 + \sigma^2 \right) = \rho^2 \left( n\|\nabla f(\theta^t)\|^2 + n\sigma^2 \right).$$

Therefore

$$\mathbb{E}\|\widehat{\mathbf{g}}^{(t)}\|^2 \leq 2\left( \|\nabla f(\theta^t)\|^2 + \frac{\sigma^2}{n} \right) + 2\rho^2 \left( n\|\nabla f(\theta^t)\|^2 + n\sigma^2 \right),$$

and plugging into step (1) gives

$$\mathbb{E}[f(\theta^{t+1})] \leq \mathbb{E}[f(\theta^t)] - \left( \eta - L\eta^2(1+n\rho^2) \right) \mathbb{E}\|\nabla f(\theta^t)\|^2 + L\eta^2 \left( \frac{\sigma^2}{n} + n\rho^2\sigma^2 \right).$$

Choose $\eta \leq \frac{1}{2L(1+n\rho^2)}$ so the coefficient on $\mathbb{E}\|\nabla f(\theta^t)\|^2$ is at least $\eta/2$, telescope over $t = 0, \ldots, T-1$, and divide by $T$ to obtain the claim. $\square$

## B  HYPERPARAMETERS FOR RESNET-18 ARCHITECTURE

All CIFAR-10 and CIFAR-100 experiments in Table 1 and Table 5 are conducted with standard hyperparameters (Zhuang et al., 2022; Cho et al., 2025; Rivaud et al., 2024; Wightman et al., 2021) an effective batch size of $\mathcal{B} = 512$, using 64 samples per GPU across $n = 8$ workers. The baseline configuration ($\mathcal{C} = 1$) is trained for standard 200 epochs. The ResNet experiments employ two learning rate schedules. The first is a cosine annealing schedule with $\eta_{\max} = 0.2$ and $\eta_{\min} = 0.002$, combined with a linear warm-up over the first 5% of training iterations to improve convergence stability. The second follows the multi-step linear schedule of Goyal et al. (2017), where the learning rate is reduced by a factor of 0.1 at predefined milestones. For CIFAR-10, these milestones are at 50% and 75% of the total training iterations, while for CIFAR-100 they occur at 30%, 60%, and 80%. The

ResNet experiments with ImageNet-1k use the standard training hyper parameters for ImageNet-1k with ResNet. Namely, a base learning rate of 0.1, paired with SGD along with a multi-step linear scheduler with milestones at 30%, 60% and 90%. An effective batch size of $\mathcal{B} = 256$, using 32 samples per GPU across $n = 8$ workers.

The ResNet experiments use group normalization layers instead of batch normalization layers with 2 groups across all experiments, ensuring that normalization is computed only over active parameters in the subnetwork configurations. Additionally, we adopt a modified Kaiming initialization (He et al., 2015), recalculating the fan-out based on the number of active (unmasked) output units. This adjustment prevents overestimation of activation variance that can occur with standard initialization when masking is applied.

## C    HYPERPARAMETERS FOR SWIN-T ARCHITECTURE

For the experiments on the CIFAR10 and CIFAR100 datasets, we use an effective batch size of $\mathcal{B} = 512$ across $n = 8$ workers, training with the AdamW optimizer with momentum for 400 epochs in the baseline DP setting. For configurations with higher sparsity, the training epochs are increased proportionally to ensure FLOP matching, as described in the previous section. As with ResNet-18, we adopt a cosine learning rate schedule with linear warm-up over the first 5% of iterations, with a peak learning rate of $\eta_{\max} = 0.0002$ and a minimum learning rate of $\eta_{\min}$ tending to 0. The experiments carried out on ImageNet use an effective batch size of $\mathcal{B} = 1024$ across $n = 8$ workers, along with an AdamW optimizer paired with a weight decay of 0.05 and a cosine annealing scheduler with $\eta_{\max} = 0.001$ and $\eta_{\min}$ tending towards zero. A linear warmup is also applied to the learning rate scheduler for the first 6.67% of epochs. In the case of $\mathcal{A} = 1$, the first 20 epochs out of 300 epochs are used as linear warm up.

## D    HYPERPARAMETERS FOR LLM ARCHITECTURE

We evaluated SDP on a 134M LLaMA-style model (Grattafiori et al., 2024), trained with a 3B-token budget (according to the Chinchilla scaling laws (Hoffmann et al., 2022) for the DDP baseline) on the FineWeb dataset. Training is performed with 7 workers and a global batch size of $920K$ tokens (sequence length 2048), using the LLaMA-2 tokenizer with a $32K$ vocabulary. Optimization follows AdamW with learning rate 8e-4 and a fixed weight decay of 0.1, and a cosine learning rate schedule with 10% warmup (from 10% of peak LR). As in other FLOP-matched settings, the number of steps in the cosine scheduler is extended accordingly. We note as well that due to compute constraints the learning rate used for all settings has been tuned for the DDP case and thus in practice SDP can potentially perform better.

## E    RESNET-18 SDP WITH MULTI-STEP LINEAR SCHEDULER

Table 5 presents the results comparing block masking and neuron masking when using a linear multi-step scheduler. We observe consistently superior performance with the **N-SDP**, especially at higher sparsity. For example, on CIFAR-100 with ResNet-18 and **N-SDP** at a coverage ratio of $\mathcal{C} = 4/8$, the accuracy achieved with linear scheduling is 58.34%, whereas **B-SDP** yields a significant degradation, reaching 40.30%. Additionally, we find that the cosine scheduler delivers even higher performance at the same coverage for both 1x and 2x model sizes. These observations demonstrate that the effectiveness of the masking techniques is robust across different learning rate schedules and architectures, underscoring their scheduler-agnostic nature.

Table 5: Top-1 test accuracy (%) with **RN-18 and WRN-18** using a multi-step linear scheduler across different coverage ratios ($\mathcal{C}$) comparing **N-SDP**, **B-SDP**, and **B$_b$-SDP** with standard DDP ($\mathcal{C}=1$). Blue cells match or exceed DDP within error bars, while at extreme sparsity ($\mathcal{C}=3/8$) **B$_b$-SDP** avoids performance collapse.

| | | **ResNet-18 (RN-18)** | | | | | |
|---|---|---|---|---|---|---|---|
| **Dataset** | **Masking** | **DDP** ($\mathcal{C}=1$) | $\mathcal{C}=7/8$ | $\mathcal{C}=6/8$ | $\mathcal{C}=5/8$ | $\mathcal{C}=4/8$ | $\mathcal{C}=3/8$ |
| CIFAR-10 | **N-SDP** | 92.41 ±0.08 | 93.14 ±0.28 | 92.95 ±0.24 | 92.23 ±0.24 | 91.25 ±0.26 | 80.93 ±2.87 |
| | **B-SDP** | | **93.18** ±0.13 | 92.64 ±0.32 | 90.35 ±0.13 | 84.01 ±0.95 | 39.35 ±0.55 |
| | **B$_b$-SDP** | | 91.62 ±0.13 | 91.50 ±0.37 | 89.61 ±0.23 | 87.94 ±0.53 | 82.18 ±0.43 |
| CIFAR-100 | **N-SDP** | 65.02 ±0.16 | 65.64 ±0.48 | 65.76 ±0.82 | 64.95 ±0.45 | 58.34 ±1.57 | 50.00 ±0.09 |
| | **B-SDP** | | **67.56** ±0.47 | 65.81 ±0.14 | 56.52 ±1.84 | 40.30 ±0.47 | 8.39 ±1.25 |
| | **B$_b$-SDP** | | 61.80 ±0.24 | 60.33 ±0.37 | 58.08 ±0.06 | 55.30 ±0.71 | 55.16 ±1.03 |
| | | **WideResNet-18 (WRN-18)** | | | | | |
| **Dataset** | **Masking** | **DDP** ($\mathcal{C}=1$) | $\mathcal{C}=7/8$ | $\mathcal{C}=6/8$ | $\mathcal{C}=5/8$ | $\mathcal{C}=4/8$ | $\mathcal{C}=3/8$ |
| CIFAR-10 | **N-SDP** | 92.26 ±0.55 | 93.47 ±0.37 | **93.97** ±0.10 | 93.74 ±0.14 | 92.51 ±0.07 | 88.23 ±2.19 |
| | **B-SDP** | | 93.66 ±0.25 | 93.28 ±0.11 | 91.19 ±0.24 | 86.90 ±0.99 | 41.11 ±1.03 |
| | **B$_b$-SDP** | | 91.95 ±0.93 | 91.79 ±0.25 | 90.04 ±0.19 | 89.20 ±0.11 | 86.20 ±0.38 |
| CIFAR-100 | **N-SDP** | 69.19 ±0.09 | **69.86** ±0.27 | 68.91 ±0.21 | 66.20 ±0.21 | 63.71 ±0.63 | 58.02 ±0.59 |
| | **B-SDP** | | 69.26 ±0.42 | 68.04 ±0.10 | 59.44 ±1.01 | 44.82 ±1.56 | 6.94 ±0.95 |
| | **B$_b$-SDP** | | 66.93 ±0.12 | 64.44 ±0.10 | 62.27 ±0.59 | 58.52 ±0.35 | 55.51 ±0.20 |

