# OpenReview forum: "Model Parallelism With Subnetwork Data Parallelism"
_ICLR.cc/2026/Conference — ICLR 2026 Conference Withdrawn Submission_

### Official Review · Reviewer_FJ4d · 2025-10-30

**Soundness:** 3
**Presentation:** 3
**Contribution:** 2
**Rating:** 6
**Confidence:** 4

**Summary:**

This paper introduces Subnetwork Data Parallelism (SDP), a distributed training framework designed
to reduce per-device memory requirements without sacrificing model quality. The key idea is to train
structured subnetworks (each a subset of the full model can be trained independently) across multiple
workers, with overlapping parameters synchronized. Two masking regimes are explored: (1) Forward
masking, where subnetworks omit parameters during both forward and backward passes, yielding
strong memory savings and implicit regularization, and (2) Backward masking, which applies sparsity
only in the backward pass, maintaining unbiased gradients and providing a theoretical foundation. The
authors present both neuron-level and block-level subnetwork construction strategies, with
experiments across CNNs, vision transformers, and LLM pretraining.

**Strengths:**

1. SDP proposes a new perspective: training overlapping subnetworks that maintain full paths from
input to loss, avoiding activation exchange. This is an elegant idea that could inspire new hybridparallel
designs.

2. The authors test across architectures (ResNet, Swin-T, LLaMA-style transformers) and datasets
(CIFAR-10/100, ImageNet, FineWeb). The breadth of experiments strengthens the paper’s
credibility.

**Weaknesses:**

1. The forward masking variant, which empirically performs best, lacks theoretical backing. This weakens the conceptual balance between theory and practice.

2. As an experiment-focus paper, the paper lacks in comparison with important baselines. For example, the results of directly applying Dropout [1] and the stochastic depth [2] are not included, which are the inspirations of the two variants of SDP.

3. In the experiments, the training epochs are extended inversely with C to have a FLOP-match comparison. It's good to keep the FLOP same for the comparison while it also increases the number of scanning on the dataset and eliminates the advantages in communication costs, which does not makes SDP superior than DDP in all aspects. It would be nice to include the results with same number of iterations.

4. The idea of SDP is similar to build the regularization mechanism into the training. It is expected to have moderate improvement in the generalization because the most experiments are performed with over-parameterized models. This limitation should be mentioned and, as mentioned in previous point, a comparison with Dropout and Stochastic Depth is important.

5. For the ResNet and Swin-transformer experiments, I think there are potential straggler problems because the computational blocks are not in an equal size and the blocks work on inputs with different sizes, which makes the computation and communication workloads imbalanced across workers. With a static block mask, Bb − SDP may suffer from the stragglers and have suboptimal runtime performance. Runtime performance should also be included to have a full picture about the contributions of SDP.

6. The estimation for the saved memory is inaccurate. As mentioned in the previous point, the computational blocks are not in an equal size, which makes the estimation based on C not accurate. Moreover, taking the example of ResNet-18, there are other layers besides the 8 computational blocks (like the first convolution layer and the fully-connected layer at the end). I think it's necessary to report the actual memory usage and the masks used in the experiments because memory saving is the main contribution.

References:
[1] Nitish Srivastava, Geoffrey Hinton, Alex Krizhevsky, Ilya Sutskever, and Ruslan Salakhutdinov.
Dropout: a simple way to prevent neural networks from overfitting. The journal of machine learning
research, 15(1):1929–1958, 2014.
[2] Gao Huang, Yu Sun, Zhuang Liu, Daniel Sedra, and Kilian Q Weinberger. Deep networks with
stochastic depth. In Computer Vision–ECCV 2016: 14th European Conference, Amsterdam, The
Netherlands, October 11–14, 2016, Proceedings, Part IV 14, pp. 646–661. Springer, 2016.

**Questions:**

1. The forward masking variant, which empirically performs best, lacks theoretical backing. This weakens the conceptual balance between theory and practice.

2. As an experiment-focus paper, the paper lacks in comparison with important baselines. For example, the results of directly applying Dropout [1] and the stochastic depth [2] are not included, which are the inspirations of the two variants of SDP.

3. In the experiments, the training epochs are extended inversely with C to have a FLOP-match comparison. It's good to keep the FLOP same for the comparison while it also increases the number of scanning on the dataset and eliminates the advantages in communication costs, which does not makes SDP superior than DDP in all aspects. It would be nice to include the results with same number of iterations.

4. The idea of SDP is similar to build the regularization mechanism into the training. It is expected to have moderate improvement in the generalization because the most experiments are performed with over-parameterized models. This limitation should be mentioned and, as mentioned in previous point, a comparison with Dropout and Stochastic Depth is important.

5. For the ResNet and Swin-transformer experiments, I think there are potential straggler problems because the computational blocks are not in an equal size and the blocks work on inputs with different sizes, which makes the computation and communication workloads imbalanced across workers. With a static block mask, Bb − SDP may suffer from the stragglers and have suboptimal runtime performance. Runtime performance should also be included to have a full picture about the contributions of SDP.

6. The estimation for the saved memory is inaccurate. As mentioned in the previous point, the computational blocks are not in an equal size, which makes the estimation based on C not accurate. Moreover, taking the example of ResNet-18, there are other layers besides the 8 computational blocks (like the first convolution layer and the fully-connected layer at the end). I think it's necessary to report the actual memory usage and the masks used in the experiments because memory saving is the main contribution.

References:
[1] Nitish Srivastava, Geoffrey Hinton, Alex Krizhevsky, Ilya Sutskever, and Ruslan Salakhutdinov.
Dropout: a simple way to prevent neural networks from overfitting. The journal of machine learning
research, 15(1):1929–1958, 2014.
[2] Gao Huang, Yu Sun, Zhuang Liu, Daniel Sedra, and Kilian Q Weinberger. Deep networks with
stochastic depth. In Computer Vision–ECCV 2016: 14th European Conference, Amsterdam, The
Netherlands, October 11–14, 2016, Proceedings, Part IV 14, pp. 646–661. Springer, 2016.

---

### Official Review · Reviewer_eWLW · 2025-10-31

**Soundness:** 2
**Presentation:** 3
**Contribution:** 2
**Rating:** 4
**Confidence:** 3

**Summary:**

Thank you for the opportunity to review this paper. This paper proposes Subnetwork Data Parallelism (SDP), a distributed training framework that reduces memory and communication costs by assigning each GPU a structured subnetwork of the full model, eliminating the need for activation exchange. Two masking regimes are explored—forward masking (i.e. sparsifies both forward and backward passes) and backward masking (i.e. applies sparsity only during backpropagation). Theoretical analysis links convergence to a spectral gap property. Experiments on CNNs, transformers, and LLMs show that SDP achieves 30–75% memory savings while maintaining or even improving accuracy compared to standard data parallelism.

**Strengths:**

1. The proposed method seems general to all model architecture.
2. The paper provides convergence guarantees to backward masking.
3.  Experiments shows significant improvement on the memory.

**Weaknesses:**

1. The theoretical investigations on the communication costs and precision loss are missing.
2. Insufficient comparison methods are compared with the proposed methods.
3. Extensive experiments on different architectures, model size, and efficiency analysis are missing, making the paper unconvincing.

**Questions:**

1. The paper does not clearly quantify the communication overhead or potential precision degradation introduced by the masking strategy.

2.  Although the title suggests a focus on model parallelism, it remains unclear how the proposed method partitions models across workers in practice, and whether it faces issues such as gradient vanishing when applied to very deep networks.

3. The experimental evaluation lacks comparisons with state-of-the-art distributed training methods, making the results less convincing. Additional experiments on larger-scale LLM architectures, with metrics on speedup and communication efficiency, would strengthen the empirical validation.

---

### Official Review · Reviewer_TDwN · 2025-11-01

**Soundness:** 2
**Presentation:** 4
**Contribution:** 3
**Rating:** 4
**Confidence:** 4

**Summary:**

This paper proposes a distributed data-parallel training framework called Subnetwork Data Parallelism (SDP), in which each worker trains on a subset of model parameters while still performing a full end-to-end forward and backward pass within its assigned subnetwork. By storing and updating only a fraction of the total parameters per worker, SDP reduces memory requirements for parameters, gradients, and optimizer states. Communication overhead is also lowered, as synchronization occurs only among workers that share overlapping parameters.

The paper introduces two complementary masking schemes for assigning parameters to workers: backward masking, where the forward pass uses the full model but masking is applied only during gradient aggregation (retaining unbiased gradients), and forward masking, where the same subset is used throughout forward, backward, and update steps (providing stronger memory savings). The authors further describe two structured subnetwork constructions: N-SDP (neuron-level masking for MLP or CNN layers) and B-SDP (block-level masking for residual or transformer blocks).

Experiments on ResNet-18, Swin-T (Tiny), and a 134M-parameter LLaMA-style model show that SDP achieves comparable (or occasionally superior) performance to standard DDP training, while reducing per-device memory usage by up to 70%. Performance remains close to baseline when the coverage ratio (fraction of parameters per worker) is moderate (>30%), but degrades at more extreme sparsity levels.

**Strengths:**

1. **Original formulation of distributed training**. The paper proposes subnetwork data parallelism (SDP), a novel formulation that bridges data and model parallelism by training structured subnetworks independently and synchronizing only overlapping parameters. This framing is original and relevant for memory-limited distributed training.
2. **Comprehensive experimental coverage**. The method is evaluated across diverse architectures (ResNet, SwinT, and LLaMA) showing consistent performance trends and generality across both vision and language domains.
3. **Clear theoretical grounding and well-structured presentation**. The analysis links convergence to the spectral properties of the masking operator, offering a solid theoretical basis. The paper is clearly written and organized, making the technical and empirical contributions easy to follow.

**Weaknesses:**

1. **Lack of empirical evidence on actual efficiency gains**. While the paper claims substantial memory and communication reduction, it does not provide quantitative measurements of actual GPU memory usage or training wall-clock time. This makes it difficult to verify the claimed advantages. Even if activation memory dominates at a per-GPU batch size of 64, the authors could still report memory statistics or time comparisons (forward, backward, and synchronization) against DDP for fairness. Without these, it is unclear how the proposed method translates into real-world efficiency.
2. **Limited applicability to very large-scale models**. The approach assumes that each worker can execute a full forward/backward pass within its subnetwork, which implicitly requires that a significant portion of the model still fits in memory (14CN = 3.5N bytes per GPU for fp32 training even without activations at 25% coverage). This constraint may limit the scalability of SDP to truly large models where model parallelism is required. The paper would benefit from discussing or demonstrating how SDP could integrate with hybrid parallelism (e.g., tensor/pipeline parallel) for billion-parameter models.

**Questions:**

1. In Section 2, SDP differs from dropout-based subnetwork training since the masking is structured and fixed, which enables efficient training. Could the authors report measured GPU memory usage and per-step runtime to substantiate the claimed efficiency gains over DDP? Such evidence could strengthen the empirical validity of the paper.
2. How are the maskings constructed? Are they random or deterministic, and what guarantees that the resulting worker-parameter graph is connected?

---

### Official Review · Reviewer_xhB7 · 2025-11-02

**Soundness:** 2
**Presentation:** 4
**Contribution:** 3
**Rating:** 2
**Confidence:** 4

**Summary:**

The paper introduces a framework for and implementation of a variant of data-parallelism during training where each worker contains only a subnetwork of the entire network being trained. This enables reduced memory requirements. A general mask-based formulation is presented for this; notably, it enables overlap in the partitions among workers. In the realization of this framework, masking can be done at a neuron or block level, and either for forward and backpropagation or only backpropagation (in which case the gradients remain unbiased). The paper develops associated theory for convergence, includes estimates for communication cost, and conducts experiments on a variety of networks/datasets/tasks.

**Strengths:**

1. The paper is tackling an important practical topic (improving the memory efficiency of training).
2. The paper is clear and very well-written.
3. There are theoretical derivations showing convergence for the backward-masking case (and the paper notes they do not have this guarantee for forward-masking).
4. The experimental evaluation covers multiple modalities (image classification, LLMs), models (ResNets, Swin, and Llama), and datasets (CIFAR-10/100, ImageNet, FineWeb).
5. The incorporation of overlap into the subnetwork partitioning is, to my knowledge, novel, and interesting.

**Weaknesses:**

Systems: Overall, the paper's claims for systems impact (e.g., reduced memory) are qualitative, not quantitative. I would like to see additional empirical measurements and baselines to supplement the discussion.
1. The paper would benefit from a detailed runtime performance analysis. Experimentally (e.g., using PyTorch's memory analysis tools), how much memory is used in each run? What is the runtime (e.g., time per mini-batch and a plot of training curve versus time)? What is the throughput (flop/s and/or device utilization)?
2. The discussion of communication focuses on bandwidth. However, this method will result in additional allreduces, each over less data and a smaller group of processes, depending on the mask. While there is less data communicated, this may result in less efficient communication. I would like to see additional analysis on this point, and experimental measurements of communication volume and efficiency.
3. The only baseline compared with is distributed data-parallelism. As the paper notes, there are other methods to reducing memory usage (sharding, model-parallelism, etc.) with different tradeoffs, especially regarding communication. However, there is no comparison with such methods. At the least, I would like to see a baseline using fully-sharded data parallelism, and ideally also DDP with checkpointing.
4. In addition to the above, it would be valuable for the paper to study the effect of the underlying hardware and network in the performance. E.g., how does throughput depend on peak network bandwidth? How does the method scale as the number of GPUs grows?
5. None of the models evaluated are especially large; all of them are likely to fit comfortably in memory for training with relatively recent GPUs. Thus, the paper is not making the empirical case that it benefits training in a memory-constrained regime. The paper would benefit from evaluating cases with larger models where memory pressure would be more acute (Llama 3 70B or 405B may be good targets here).

Experimental results:
1. Tables in the paper report error bars, but it is not clear how they were calculated. How many runs were used? How were the error bars calculated?
2. The results consistently show that B-SDP achieves the best performance, despite B$_\beta$-SDP having stronger theoretical justification and empirically more similar gradients. I would have liked to see more in-depth analysis of why this is.
3. It is not clear whether the method scales with larger models/datasets. The ImageNet results fail to match the baseline, and the LLM results use only a tiny model (134M parameters), raising the concern that for a larger LLM, the method will also fail to match the baseline. A scaling study as model parameters grow for the LLM would be a valuable addition.
4. Mixture-of-experts would seem to have many similar memory benefits to the proposed approach (though there is obviously no overlap). An additional baseline comparing with MoE would strengthen the case made in the paper.

Theory:
1. There is very little theory for the forward-masking case, which is unfortunate as it is the version that seems to perform best in practice. E.g., can you bound the bias in the gradients here?
2. The spectral gap $\rho$ is a critical constant for many bounds. Can you calculate it for the masking regimes you use (e.g., depending on the overlap), or otherwise bound it?

**Questions:**

Please see the weaknesses above for more details. Ideally, all the above issues would be addressed. Below are some key questions; I would be willing to raise my score to at least borderline if most of these were answered well.

1. (See Systems-1): What is the experimental memory use, runtime, and throughput?
2. (See Systems-3): How does the method compare against FSDP and DDP with checkpointing?
3. (See Experimental-1): How many runs were used for error bars, and how were they calculated?
4. (See Experimental-3 and Systems-5): How does the method perform, both in runtime/memory and perplexity, for larger models?
5. (See Theory): Can you provide a stronger theoretical grounding for the forward-masking case? What can you say about $\rho$?

---

### Note · Authors · 2025-11-23

**Comment:**

Dear Reviewers,

Thank you for your feedback which helps us improve the paper. We have decided to withdraw the paper and resubmit in a future conference to allow more time to enhance the writing and obtain additional experiments addressing the reviewers points.

**Withdrawal Confirmation:**

I have read and agree with the venue's withdrawal policy on behalf of myself and my co-authors.